# Investigation on the Mass Distribution and Chemical Compositions of Various Ionic Liquids-Extracted Coal Fragments and Their Effects on the Electrochemical Performance of Coal-Derived Carbon Nanofibers (CCNFs)

**DOI:** 10.3390/nano11030664

**Published:** 2021-03-08

**Authors:** Shuai Tan, Theodore John Kraus, Mitchell Ross Helling, Rudolph Kurtzer Mignon, Franco Basile, Katie Dongmei Li-Oakey

**Affiliations:** 1Department of Chemical Engineering, University of Wyoming, 1000 E University Ave, Laramie, WY 82071, USA; shuaitan.che@gmail.com; 2Department of Chemistry, University of Wyoming, 1000 E University Ave, Laramie, WY 82071, USA; Kraus.ted@gmail.com (T.J.K.); mhelling@uwyo.edu (M.R.H.); rudolphmignon@gmail.com (R.K.M.); Basile@uwyo.edu (F.B.)

**Keywords:** coal, ionic liquids, electrospinning, carbon nanofibers, supercapacitors

## Abstract

Coal-derived carbon nanofibers (CCNFs) have been recently found to be a promising and low-cost electrode material for high-performance supercapacitors. However, the knowledge gap still exists between holistic understanding of coal precursors derived from different solvents and resulting CCNFs’ properties, prohibiting further optimization of their electrochemical performance. In this paper, assisted by laser desorption/ionization (LDI) and gas chromatography–mass spectrometry (GC–MS) technologies, a systematic study was performed to holistically characterize mass distribution and chemical composition of coal precursors derived from various ionic liquids (ILs) as extractants. Sequentially, X-ray photoelectron spectroscopy (XPS) revealed that the differences in chemical properties of various coal products significantly affected the surface oxygen concentrations and certain species distributions on the CCNFs, which, in turn, determined the electrochemical performances of CCNFs as electrode materials. We report that the CCNF that was produced by an oxygen-rich coal fragment from C_6_mimCl ionic liquid extraction showed the highest concentrations of quinone and ester groups on the surface. Consequentially, C_6_mimCl-CCNF achieved the highest specific capacitance and lowest ion diffusion resistance. Finally, a symmetric carbon/carbon supercapacitor fabricated with such CCNF as electrode delivered an energy density of 21.1 Wh/kg at the power density of 0.6 kW/kg, which is comparable to commercial active carbon supercapacitors.

## 1. Introduction

Thanks to their excellent conductivity, chemical stability and highly porous structure, carbon nanofibers (CNFs) have been widely used as electrode materials in high-performance supercapacitors (SCs) [1,2,3,4]. Compared to other carbon-based materials, such as active carbons (ACs) [5], carbon nanotubes (CNTs) and graphene [6], CNFs allow for more flexible electronic components that are more suitable for practical applications [7,8,9,10,11,12]. Electrospinning is an economical and scalable process, which has been applied to fabricate CNFs often from spinnable solutions, such as polyacrylonitrile (PAN), polyvinylpyrrolidone (PVP) and polyvinyl alcohol (PVA) [4,13,14,15,16,17]. In order to achieve superior energy density or power density, these polymer-based CNFs can be modified by adding precursors, such as metal oxides [18,19,20,21], nitrogen-containing materials [22,23,24,25,26,27,28,29,30], or further functionalized into two-dimensional or even three-dimensional structures [31,32,33]. However, these modifications require high-cost materials and complex fabrication processes, which are the main roadblocks for practical application. As such, it is essential to use alternatively economic materials to fabricate CNFs with flexible structure and desirable chemical properties. In this respect, coal derivatives, referred here to as coal tar pitch and/or coal char derived from facile, low energy, solvent extraction processes, have been found to be a cheap and abundant material in fabricating CNFs and other structural materials, such as hierarchical porous carbon and carbon nanosheets [34,35,36,37]. Because of the high degree of graphitization and decomposition of coal precursors during the carbonization process, highly conductive and porous CNFs have been achieved via electrospinning process. Importantly, the nitrogen and oxygen functional groups in coal precursors could significantly increase the energy density of resultant CNFs, thanks to pseudocapacitance, induced by faradic reactions between those functional groups, and the electrolyte [38,39].

Traditionally, raw coal was depolymerized by using organic solvents, such as hexane [40], tetrahydrofuran (THF) [41] and *N*-Methyl-2-pyrrolidone (NMP) [42,43,44], or strong acids (H_2_SO_4_/HNO_3_) to obtain coal fragments, which led to a process that is not environmentally friendly, thus limiting the scaleup of CNFs’ fabrication. In contrast, ionic liquids (ILs) as an environmentally friendly solvent with negligible volatility and corrosiveness have been reported to be effective in coal depolymerization [45,46] and biomass extraction [47,48,49] under mild conditions (<100 °C). Our group recently reported the feasibility of applying coal char derived from 1-Butyl-3-methylimidazolium chloride (C_4_mimCl) extraction into coal-derived carbon nanofibers (CCNFs) via an electrospinning process [38,39]. A specific capacitance of 299.4 F/g at a current density of 1 A/g was achieved with CCNF mats as supercapacitor electrodes, demonstrating 93% capacitance retention after 10,000 cycles of galvanostatic charge–discharge (GCD). Most importantly, our following study found that, even though the yield of coal fragments derived from ILs extraction (15–30%) was lower than the one obtained from traditional organic solvents (e.g., 50% using tetralin), the ILs, such as 1-butyl-2,3-dimethylimidazolium chloride (C_4_m_2_imCl), can have strong selectivity toward certain groups of fragments in coal extraction, such as a preference to both polycyclic aromatic hydrocarbons (PAHs) and fatty acids (FAs) [50]. Therefore, the selectivity in extracting coal compounds is a unique feature of ILs, and it is worth deeply understanding the role of chemical properties of ILs-derived coal precursors in the electrochemical performances of resulting CCNFs. In this paper, we report that coal fragments derived from two types of coal (subbituminous and bituminous) and four different ILs were systemically characterized by laser desorption/ionization (LDI) and gas chromatography–mass spectrometry (GC–MS). Correspondingly, the electrochemical performance of the resulting CCNFs from respective ILs was characterized and compared to distinguish the effects of mass distribution and functional groups from each IL extractant. By characterizing the homogeneity of as-spun fibers and surface chemistry of resulting CCNFs, we demonstrated that the electrochemical performances of CCNFs were significantly affected by the mass distributions and compositions of chemical compounds in the IL-extracted coal precursors. It should be noted that the surface area and volume of meso- and micropores were found to be similar among the different CCNFs. Therefore, in this manuscript, we mainly focused on the difference on the surface chemistry in each CCNF. We hope that this study can provide guidance toward precursor processing and selection in CNF manufacturing.

## 2. Materials and Methods

### 2.1. Materials

Poly (acrylonitrile-co-methyl acrylate) (PANMA) (acrylonitrile weight percentage: ~94%) and polyvinylpyrrolidone (PVP) (360,000 g/mol) were purchased from Sigma–Aldrich (St. Louis, MO, USA). PANMA acted as C- and N- source precursor, and PVP acted as a sacrificial material. Standard Reference Material^®^ 2682c subbituminous coal (referred as 2682 coal) from Wyodak–Anderson coal seam (powder river basin, PRB Coal), and Standard Reference Materials^®^ 2684c bituminous coal (referred as 2684 coal) from Pittsburgh coal seam were purchased from National Institution of Standards and Technology (NIST) (Gaithersburg, MD, USA). Figure 1 lists four ILs (>98.0% HPLC and <1.0% H_2_O impurity) used in coal extraction: 1-butyl-2,3-dimethylimidazolium chloride (C_4_m_2_imCl), 1-ethyl-3-methylimidazolium chloride (C_2_mimCl), 1-Hexyl-3-methylimidazolium chloride (C_6_mimCl) and 1-Butyl-3-methylimidazolium chloride (C_4_mimCl), which were purchased from Sigma–Aldrich (St. Louis, MO, USA) and used as extraction solvents without further purification. 1-Methyl-2-pyrrolidinone (NMP) (99.5%) (Sigma–Aldrich) was used as a solvent to obtain coal char from coal/ILs solution through a centrifuge. *N*,*N*-Dimethylformamide (DMF, 99.8%) (Sigma–Aldrich) was used as an organic solvent to dissolve PANMA, PVP and coal precursors for electrospinning process.

### 2.2. Characterizations of Coal Precursors

The experimental process of obtaining coal chars from ILs extraction was described in our recently published papers [38,39]. Briefly, 1 g of coal was mixed with 10 g of ILs in a test tube, which was then heated at 100 °C overnight in an oil bath. The resulting viscous mixture of coal and IL was cooled to room temperature, followed by centrifugation at 4000 rpm for more than 30 min to obtain coal char. Coal char was then washed with deionized (D.I.) water in a Soxhlet extractor to remove ILs and NMP, followed by drying in a vacuum desiccator at room temperature. Dried coal char was further dissolved in DMF at 80 °C overnight under constant stirring, followed by centrifugation to obtain soluble coal char in DMF. Since trace amounts of ILs still remained in the coal char/DMF solution, an additional purification step to remove the trace ILs was carried out by following the procedures reported in open literature [50]. Sequentially, the purified coal char/DMF solution was analyzed by laser desorption and ionization–mass spectrometry (LDI-MS) (5800™, Sciex, Redwood City, CA, USA). Specifically, 1 µL of polyethylene glycol 600 (PEG 600) was deposited onto a sample plate for external calibration purpose. Then, 1 µL clean sample was deposited onto the plate and dried under atmospheric conditions. All the mass spectra were recorded in reflector mode with an accelerating voltage of 15 kV and a delay duration of 100 ns. Each mass spectrum was collected in the 10 to 800 *m/z* range and was the result of averaging 400 laser shots at 4500 laser intensity (arbitrary units; a.u.). The mass spectral data were analyzed by using DataExplorer (Sciex) and the mMass open source software (www.mmass.org, accessed on 26 February 2021).

GC–MS analysis of coal chars was carried out by a gas chromatography (GC) (Trace GC Ultra; Thermo, Salt Lake City, UT, USA) coupled to a single quadrupole mass spectrometry (MS) system (Model DSQ; Thermo). The specific procedures are reported in detail in our previous publication [50].

### 2.3. CCNFs Fabrication

Certain amounts of PVP and PANMA were added into coal precursor/DMF solution to obtain 5 wt% PVP/5 wt% PANMA/0.5 wt% coal precursor in DMF. Sequentially, the electrospinning process was carried out in EC-DIG+ electrospinner (IME Medical Technology). Specifically, a spinnable solution was filled into a 5 mL syringe associated with 0.8 mm OD nozzle. Controlled by a syringe pump, the solution was delivered at a flowrate of 20 µL/minute and electrospun at the voltage of 16 kV onto a rotated drum, which was charged at −2 kV. The distance between the nozzle and the drum was kept at 18 cm to maintain 1 kV/cm electronic field. Relative humidity and temperature in the electrospinning fume hood were maintained at 30% and 20 °C, respectively. The electrospun fiber mat was peeled off from a drum using metallic clips. In order to provide constant stress on the mat that is below the breaking point, the mat was stretched to 1.2 times the original fiber length. The fiber mat was stabilized in an oven at 300 °C (1 °C/min) for 3 h under air, then transferred into a tube furnace to be carbonized at 1000 °C at the heating of 5 °C/min for 1 h under nitrogen (N_2_). The stabilization and carbonization processes were also characterized by thermogravimetric analysis (TGA) using some parameters to study the yield of CCNFs for each coal char precursor. The result was summarized in Appendix A. Specifically, in the oxidation step (room temperature to 300 °C), nearly 5 wt% water adsorbed in the as-spun fibers was removed in the early stage (<100 °C), followed by a 40 wt% mass loss from 200 °C to 300 °C. Such weight loss was caused by the decomposition of PVP as sacrificial agent to create the porous structure and exothermic process of PAN cyclization. Sequentially, in the carbonization process, carbon ring condensation and breakage of oxygen and nitrogen functional groups occurred, which usually accompanied with the release of CH_4_, CO_2_ and CO. Approximately 25 wt% weight loss was observed between 400 °C and 1000 °C in the carbonization process, which generated CCNFs with oxygen residues in the turbostratic carbon structure.

### 2.4. Characterizations of CCNFs

The surface chemistries of fabricated CCNFs were characterized by Kratos Ultra DLD X-ray photoelectron spectroscopy (XPS) (Manchester, UK) with a monochromatic Al-Kα x-ray source under 10^−10^ torr vacuum. Survey scans were performed at 80 eV pass energy with an energy resolution of 0.5 eV and three sweeps. High-resolution scans of C 1s, N1s and O1s were performed at 40 eV pass energy with an energy resolution of 0.02 eV and three sweeps.

### 2.5. Electrochemical Measurements

The electrochemical properties of CCNFs were measured in 6 mol/L KOH using a three-electrode cell, which consisted of a 1 cm × 1 cm carbon fiber mat that was pressed between nickel foams on both sides as the working electrode, Pt foil as a counter electrode and Ag/AgCl as a reference electrode. Cyclic voltammetry (CV) profiles were recorded from −1.1 V to −0.1 V at different scan rates (5–15 mV/s). Galvanostatic charge–discharge (GCD) was performed at the same potential range (−1.1 V to −0.1 V) by varying the current densities from 1–5 A/g, which are used to calculate the specific capacitances of CCNFs by Equation (1):(1)$$C_{s}=I_{D}\times \frac{\Delta t}{\Delta V}$$
where C_s_ is the specific capacitance of the electrode in Farads per gram (F/g), I_D_ is the discharge current density in ampere per gram (A/g), Δt is the discharge time in second (s) and ΔV is the potential range in volts (V). Electrochemical impedance spectroscopy (EIS) was measured with an amplitude of 10 mV in the frequency range from 10 kHz to 0.01 Hz.

Carbon/carbon supercapacitor cells consist of (1) one 4 cm × 3 cm Lucite piece as cell support, (2) nickel foam as the current collector and (3) 1 cm × 1 cm CCNFs mat as the electrode on each side, which was separated by Ryton current separator. An entire cell was immersed in 6 M KOH electrolyte to measure electrochemical properties in a two-electrode cell mode. Power and energy densities of a carbon/carbon supercapacitor cell are calculated by following Equations (2) and (3):(2)$$P=\frac{1}{2}\times I\times \Delta V$$
(3)$$E=\frac{1}{2}\times C\times {\left(\Delta V\right)}^{2}$$
where ΔV is the sweep potential window (V), I is the applied current density (A/g) and C is calculated specific capacitance (F/g), calculated from Equation (1).

## 3. Results

### 3.1. Chemical Properties of ILs-Derived Coal Precursors

The interaction of ILs with coal varies with different cations and anions, which can be characterized using the interaction parameters, including hydrogen bond basicity, hydrogen bond acidity, π- or n-(nonbonding) interaction and dipolarity/polarizability. According to the findings from Anderson et al. [51], the hydrogen bond basicity is largely affected by the anion species, with Cl^−^ being the anion with the highest basicity among the common species in their study, such as BF_4_^−^, PF_6_^−^. This means that the ILs associated with Cl^−^ anion have denser HOMO lone pair electrons in the highest occupied molecule orbital (HOMO) to be reoriented, facilitating hydrogen bonding with solutes [52]. As such, the functional groups (e.g., hydroxyl and carboxyl) in coal could have strong hydrogen bonding with IL Cl^−^ anion. Furthermore, another parameter, π- or n- (nonbonding) interaction, can also be strongly formed in the case of an electron-rich aromatic π-system existing in cations, such as C_4_m_2_im^+^ and C_6_mim^+^. Even though C_2_mim^+^ and C_4_mim^+^ have no analogous aromatic system, it was found that the combination of nonbonding electrons in Cl^−^ anion and these cations provides the capability to form π- or n- interaction with solutes [51].

As shown in Figure 2, the LDI mass spectra of four Cl^−^ IL (C_4_m_2_im^+^, C_2_mim^+^, C_4_mim^+^ and C_6_mim^+^)-extracted 2682 coal fragments were comprised of similar compounds but with different relative intensities. Specifically, the detected coal compounds were found to be abundant in the range from *m*/*z* 400 to *m*/*z* 430 in the C_4_m_2_imCl-2682 coal extract (Figure 2A), whereas the mass range of primary compounds shifted to *m*/*z* 330–380 in C_4_mimCl-2682 coal extract (Figure 2C) and further declined to lower mass range (*m*/*z* 150 to *m*/*z* 260) in both C_2_mimCl-2682 (Figure 2B) and C_6_mimCl-2682 coal extracts (Figure 2D). Since the LDI spectra have been externally calibrated by PEG 600, the molecular weight of each detected compound in LDI is accurate to 60 ppm (or Delta *m*/*z* +/- 0.01) and can be used to estimate their chemical compositions. As such, assisted by the Molecular Weight Calculator (Matthew Monroe; http://www.alchemistmatt.com/, version 6.50, accessed on 26 February 2021) and PubChem Open Chemistry Database, the abundant compounds at molecular weights between *m*/*z* 200 and *m*/*z* 300, such as the significant peaks at 211.05 and 268.15, were tentatively identified to be the components that were consist of carbon chains (<8) associated with 4–5 oxygen atoms. Most of the compounds above *m*/*z* 300 were tentatively identified to be aromatics (>C_13_) associated with just one carboxyl or ether group. It should be noted that a unique compound (*m*/*z* 167.01) was detected in the LDI spectra of C_6_mimCl-2682 coal precursor (Figure 2D), which may be attributed to one quinone, one ether and one carboxyl group. Although the specific functional groups may need additional analysis by a van Krevelen plot, it seems that the lighter compounds (also known as lower *m*/*z*) in coal fragments contain more oxygen atoms than the heavier compounds, which appear to be rich in carbon. Since 2684 bituminous coal has more aromatics than 2682 subbituminous coal, we hypothesize that the π- or n- interaction may become more significant in 2684-coal extraction with the methylimidazolium-based ILs in this study. Indeed, compared to the LDI spectra of C_6_mimCl-2682 coal extract (Figure 2D), the 2684 coal extract derived from the same IL presented not only much more compound species but also a mass distribution in the heavier compound regime, with detectable *m*/*z* intensities above 500 (Figure 2E). The sequential analysis by saturates–aromatics–resins–asphaltenes determinator (SAR–AD) revealed that, for the same IL (C_6_mimCl), extracts from aromatics-rich 2684 coal indeed showed four times higher concentration of asphaltenes than C_6_mimCl-2682, mainly consisting of large polyaromatic molecules (Appendix A). These results show that extract product preferences can be carefully engineered using ILs for different coal samples.

To further characterize various IL-derived coal extracts, the concentrated coal extracts were prepared for GC–MS analysis by drying a 25 µL aliquot under nitrogen. The dried extract was derivatized with 25 µL *N*-Methyl-*N*-(trimethylsilyl) trifluoroacetamide (MSTFA) for one hour at 40 °C. We used 1 µL injection for the GC–MS analysis (Figure 3). Data files were processed with AMDIS for deconvolution and mass spectral database searching (NIST 17). Deconvoluted mass spectra were considered tentatively identified when they had a database match factor of 700 or greater. Compounds from MSTFA or known contaminants (e.g., plasticizers, septa and column bleed) were omitted from further consideration. Several compounds were consistently detected among the different extracts, including methylamine (CH_3_NH_2_), ethylene glycol (C_2_H_6_O_2_), boric acid (H_3_BO_3_) and lactic acid (C_3_H_6_O_3_). Several other compounds contained carboxyl and hydroxyl groups but presented different relative abundance among the various coal extracts. It should be noted that the MS spectra of several highly abundant compounds with retention times of 12.4 min, 13.98 min and 17.35 min had low match scores (below 700) with NIST17 database and, thus, were not tentatively identified. The compounds that were tentatively identified in each coal extract are listed in Appendix A. While most of the aromatics were detected in LDI–MS analysis, GC–MS analysis mainly revealed the polar compounds. Therefore, LDI–MS and GC–MS detected complementary compounds.

To summarize, LDI–MS and GC–MS analysis showed that C_6_mimCl- and C_2_mimCl-derived coal extracts tend to have lighter compounds than those from C_4_m_2_imCl and C_4_mimCl. The aromatics-rich 2684 coal samples yielded more compounds in the extracts than 2682 (PRB coal) for the same IL, which may be attributed to the different chemical nature of 2684 bituminous and 2682 subbituminous samples. The following discussion focuses on how such chemical property differences in various IL-derived coal extracts may result in distinct surface chemistry and, consequentially, electrochemical performance, of the resulting CCNFs.

### 3.2. Surface Properties of CCNFs

According to our recent study, the introduction of coal precursors into carbon fibers could result in an oxygen-enriched surface that was responsible for enhanced wettability and pseudocapacitance [38,39]. However, there still lacks the systematic investigation on the change of surface properties, including total oxygen concentration and oxygen species distribution, of fabricated CCNFs from different IL-derived coal precursors. As summarized in Table 1, the total oxygen concentrations calculated from XPS survey scan (left column in Figure 4) showed that C_2_mimCl-2682 (surface oxygen: 4.1%) and C_6_mimCl-2682 CCNFs (surface oxygen: 4.2%) had higher surface oxygen concentrations than C_4_m_2_imCl- (1.9%) and C_4_mimCl-2682 CCNFs (2.6%). According to the LDI and GC–MS analysis, the mass distributions of abundant compounds in C_4_m_2_imCl- and C_4_mimCl- 2682 coal extracts were mostly above *m*/*z* 300, consisting of aromatics with long carbon chains (>13) but fewer oxygen atoms (1–2). In contrast, the lighter compounds (<260 Da) in C_2_mimCl- and C_6_mimCl- 2682 coal extracts had relatively shorter carbon chains (<8) associated with more oxygen atoms (4–5), which may be responsible for the higher surface oxygen concentrations in the resulting CCNFs. Moreover, due to the low oxygen contribution from the carbon-rich 2684 bituminous coal and asphaltene feature of the C_6_mimCl-2684 coal extract (Appendix A), the resulting C_6_mimCl-2684 CCNFs only contained 2.6% overall oxygen concentration, which was much lower than the 4.2% oxygen content in the C_6_mimCl-2682 CCNFs.

Aside from the overall oxygen concentration, O1s peaks were deconvoluted using Gaussian–Lorentz peak fitting. Three different O1s peaks were shown at 530.5 eV, 532 eV and 534 eV binding energies, which were assigned to C–O quinone type (O_I_), C–OH phenol and/or C–O–C ether (O_II_) and COOH carboxylic groups (O_III_), respectively. Quinone groups (O_I_) can have redox reactions with acidic electrolytes, attributing to additional pseudocapacitance, in addition to electrochemical double layer capacitance. The O_I_ concentration in C_2_mimCl- (29.5%) and C_6_mimCl- (38.1%) 2682 CCNFs were significantly higher than it in C_4_m_2_imCl- (18.9%) and C_4_mimCl- (23.8%) 2682 CCNFs. This observation is consistent with higher aromatics compounds in C_2_mimCl- (69.68%) and C_6_mimCl- 2682 (62.40%) coal extracts than with those from C_4_m_2_imCl- (46.89%) and C_4_mimCl- (58.04%) (Appendix A).

The wettability of CCNFs is another factor that facilitates the ion migration and accessibility into micropores, often resulting in better electrochemical performance. According to the literature, the wettability is closely related to the concentration of CO-type functional groups, especially the C–OH phenol groups and/or C–O–C ether groups (O_II_) [34]. Due to the instability of COOH groups (O_III_) during electrochemical cycling, it could not effectively facilitate the fiber wettability. As shown in Table 1, even though C_6_mimCl-2682 CCNF had the lowest O_II_ distribution (47.2%), it showed the highest O_II_ overall oxygen concentration (2.0%) when accounting for its total oxygen concentration (4.2%). For the same C_6_mimCl IL, the aromatics-rich 2684-coal fragment led to much lower overall oxygen concentration (2.6%) in the resulting C_6_mimCl-2684 CCNFs.

The XPS study also revealed that small amount of coal extract in electrospinning solutions, such as 0.5 wt% in total weight of polymer binder and solvent, resulted in significant changes in the surface chemistry of resulting CCNFs, which, consequently, could play a crucial role in their electrochemical properties.

### 3.3. Electrochemical Measurements

#### 3.3.1. CCNFs Derived from ILs-Extracted 2682 Coal Fragment

As shown in Figure 5A, the fabricated CCNFs presented rectangular CV profiles, signature of electrical double-layer capacitance. It should be noted that both C_2_mimCl- and C_6_mimCl-2682

CCNFs showed a significant capacitance increase in the potential range of −0.8 V to −0.5 V, compared to C_4_m_2_imCl- and C_4_mimCl- 2682 CCNFs. This increase may be attributed to pseudocapacitance of C_2_mimCl- and C_6_mimCl- 2682 CCNFs in basic electrolytes, possibly resulting from the high concentrations of oxygen functional groups, particularly carboxylate and phenol groups (Table 1). The differences in surface chemistry of various CCNFs also affected their specific capacitances (Figure 5C), which were calculated from their corresponding GCD profiles (Figure 5B). At the current density of 1A/g (Figure 5C), C_6_mimCl-2682 CCNFs achieved the highest specific capacitance (247.0 F/g), which is 15% higher than that of C_2_mimCl-2682 CCNFs (215.2 F/g), 71% higher than C_4_m_2_imCl- (144.8 F/g) and 50% than C_4_mimCl- (165.2 F/g) 2682 CCNFs. The specific capacitance of C_6_mimCl-2682 CCNFs remained as the highest (165.0 F/g) as current density increased to 20 A/g. Sequentially, electrochemical impedance spectroscopy (EIS) was carried out to evaluate the dynamic properties of electrolyte ions relative to the fabricated CCNFs. As shown in Figure 5D, a semicircle that appeared in the high-frequency region (10 KHz–10Hz) represents the internal resistance of electrodes. As mentioned with the XPS analysis, C_6_mimCl- and C_2_mimCl- 2682 CCNFs had higher O_II_ concentrations (Table 1), which could enhance the wettability of CCNFs and, consequently, reduce the internal resistance of CCNFs to facilitate the transfer of electrolyte ions. As such, C_6_mimCl- and C_2_mimCl- 2682 CCNFs presented smaller semicircles than C_4_m_2_mCl- and C_4_mimCl- 2682 CCNFs, indicating lower internal resistance values (~2 Ω) when the experimental EIS data was simulated with Z_view_ software (Table 2). Additionally, the ion diffusion into the pores of CCNFs was improved, evidenced by the lower diffusion resistance (R_p_) value shown in C_6_mimCl-2682 CCNFs (0.035 Ω) than in C_4_mimCl-2682 (0.052 Ω). Even though the micropores in CCNFs cannot be fully wetted to trap ions, enhanced wettability resulted from higher O_II_ groups in C_6_mimCl- and C_2_mimCl- 2862 CCNFs can improve the ion accessibility into the micropores, resulting in lower ion diffusion resistance.

To further study the influence of surface oxygen on the electrochemical properties of resulting CCNFs, the Trasatti method was applied to quantify the pseudocapacitance contribution. In the plot of 1/q vs. v^1/2^ (Figure 6, left), the total voltammetric charge, q_T_, is calculated when v = 0. Charge associated with the electrical double-layer, q_EDL_, is calculated when v = ∞ in the plot of q vs. v^−1/2^ (Figure 6, right). Charge associated with pseudocapacitance (q_p_) is obtained by the difference between q_T_ and q_EDL_. Finally, the corresponding maximum total specific capacitance (C_S,T,M_), electrical double layer capacitance (C_EDL_) and pseudocapacitance (C_p_) can be calculated by dividing the respective potential ranges used to collect the CV data. As summarized in Figure 7, C_6_mimCl-2682 CCNFs achieved the highest contribution of C_p_ (54.7%), while C_4_m_2_imCl-2682 CCNFs obtained only 45.3% of C_p_ contribution. The trend for C_p_ is consistent with that of the surface oxygen concentration of fabricated CCNFs (Table 1), revealing that higher surface oxygen level may be responsible for the enhancement of electrochemical performance of CCNFs in the form of increased pseudocapacitance contribution.

A carbon/carbon symmetric supercapacitor was assembled (inset of Figure 8C), in which the CCNFs were used as both anode and cathode electrodes and tested in the two-electrode mode. In order to obtain a better comparison with substantial difference on the cell performance, the concentration of coal precursor was increased to 1 wt% in the electrospinning solution. In Figure 8C, theF two-electrode cell made with C_6_mimCl-2682 CCNFs achieved the highest specific capacitance, resulting in an energy density of 21.1 Wh/kg at the power density of 0.6 kW/kg (Figure 8C).

Impressively, even at a much higher power density (12 kW/kg), the C_6_mimCl-2682 supercapacitor cell can still achieve an energy density of 7.6 Wh/kg, which was out-performed by other CCNFs derived from different coal extracts (Appendix A). Being consistent with the EIS study of CCNFs in three-electrode mode (Figure 5D), C_6_miCl-2682 supercapacitor cell presented the smallest semicircle at high frequency (Figure 8D). Additionally, the slope of straight line at low frequency (1 Hz to 0.01 Hz) tended to be most vertical in C_6_mimCl-2682 supercapacitor cell (Figure 8D), indicating the lowest ion diffusion resistance. The durability of the electrode was studied by performing 1000 cycles of GCD at the current density of 1A/g. The capacitance after every 100 cycles was recorded and summarized in Appendix A, which shows >90% capacitance retention for all CCNF electrodes.

#### 3.3.2. CCNFs Derived from an ILs-Extracted 2684 Coal Precursor

To investigate the effect of coal sample differences via extraction using the same IL (C_6_mimCl), the CCNFs from aromatics-rich 2684 bituminous coal 2684 (also known as C_6_mimCl-2684 CCNFs) showed much lower surface oxygen concentration and poorer electrochemical performances than CCNFs from subbituminous coal 2682 (also known as C6mimCl-2682 CCNFs) (Figure 9).

Specifically, the specific capacitance of C_6_mimCl-2684 CCNFs only had 135 F/g (Figure 9C) at a current density of 1A/g, much lower than that of C_6_mimCl-2682 CCNFs (247 F/g) (Figure 5C). Additionally, both internal and ion diffusion resistances in C_6_mimCl-2684 CCNFs were significantly higher than C_6_mimCl-2682 CCNFs, evidenced by a larger semicircle at the higher frequency and a line with less inclination at the low frequency in EIS measurement (Figure 9D). It should be noted that the specific capacitance of C_6_mimCl-2684 CCNFs (135 F/g) was even lower than the CNFs without adding coal precursor (157.8 F/g), indicating the introduction of C_6_mimCl-2684 coal precursor worsens the electrochemical performance of resulting CCNFs.

To probe the morphological differences between C6mimCl-2682 and C6mimCl-2684 CCNFs, the as-spun nanofibers were investigated under microscope. As demonstrated in Figure 10, the nanofibers fabricated from PANMA/PVP precursor showed homogeneous fiber appearance (Figure 10A,B). After adding the C6mimCl-2682 coal fragments’ slightly darker regions appeared in the fiber network (Figure 10C,D), indicating that the coal fragments were well distributed and that the C6mimCl-2682 coal fragments were compatible with PANMA/PVP solution. In contrast, as-spun nanofibers from adding C6mim-2684 coal fragments showed the poor homogeneity and aggregation of coal fragments (Figure 10E,F), illuminating the poor miscibility between the C6mim-2684 coal fragments and the PANMA/PVP solution. We hypothesize that the large amount of asphaltenes in bituminous C6mimCl-2684 coal fragments could not mix well with PANMA and PVP polymeric solution in the polar DMF solvent. Consequently, the aggregates of C6mimCl-2684 coal fragments in the as-spun nanofibers formed large sections with low pore volume after thermal treatment, resulting in poor ion transfer and diffusion in the final CCNFs. Therefore, the mass distribution and chemical compositions of the coal fragments require serious consideration in fiber fabrication processes since they could play a crucial role in the electrochemical properties of the final fibers.

## 4. Conclusions

In this paper, we report a systematic study on the chemical properties of coal extracts from various ionic liquids characterized using LDI and GC–MS analyses for PRB coal (subbituminous 2682), revealing the significant differences in their mass distribution and chemical compositions. Following this understanding of different IL extracts, XPS analysis of the resulting coal-derived carbon nanofibers (CCNFs) demonstrated that the chemical properties of coal extracts changed the surface chemistry of CCNFs, which, consequently, affected the electrochemical performances of CCNFs as supercapacitor electrodes. Specifically, thanks to the strong hydrogen and π-/n- interactions with inter-molecular bonds in the coal network induced by C_6_mim^+^ and Cl^−^, light coal fragments associated with oxygen functional groups were extracted from subbituminous 2682 coal. Adding this 2682 coal extract to spinning solutions showed that the C_6_mimCl-2682 CCNFs possess high concentration of surface oxygen, which may be attributed to the enhanced overall capacitance in the form of pseudocapacitance. Additionally, the high concentration of C–OH phenol and/or C–O–C ether groups in C_6_mimCl-2682 CCNFs led to enhanced wettability, evidenced by the reduced ion transfer/diffusion resistances in the EIS results. Finally, the symmetrical supercapacitor cells from C_6_mimCl-2682 CCNFs generated energy density of 21.1 Wh/kg at the power density of 0.6 kW/kg. In contrast, other ILs extracted coal fragments from subbituminous 2682 coal samples tend to possess heavier compounds with less oxygen content, resulting in subpar electrochemical performances of the final fiber mats.

To summarize, the chemical characterization of IL-extracted coal fragments, followed by sequential surface science study and electrochemical performance of the final fiber, provides a holistic approach to understand and design liquid extraction processes of coal in order to utilize this abundant and cheap feedstock resource for nonthermal applications. While we used two categories of coal in this study, our approach can be easily extended to biomasses. Similarly, CCNFs from coal extracts may be useful for electrodes in other energy storage devices, such as sodium ion batteries, in addition to the high-performance supercapacitor application reported in this paper.

## Figures and Tables

**Figure 1 nanomaterials-11-00664-f001:**
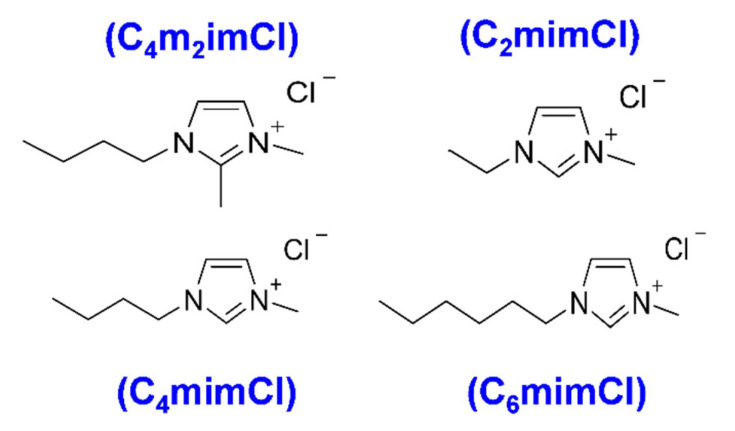
The ionic liquids (ILs), C_4_m_2_imCl, C_2_mimCl, C_4_mimCl and C_6_mimCl used in this study, and their corresponding structures.

**Figure 2 nanomaterials-11-00664-f002:**
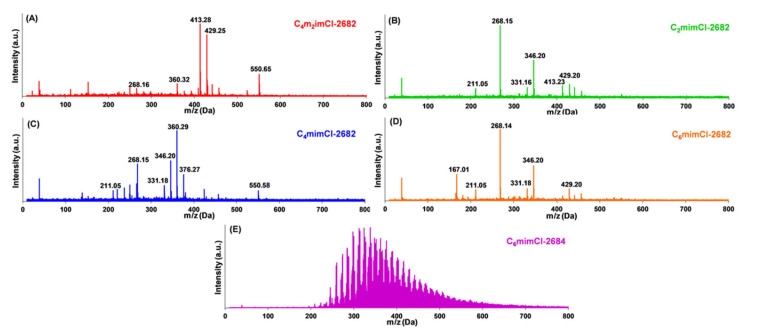
The MALDI spectra of ILs-extracted coal precursors from NIST 2682 with different ILs as extractants. (**A**) C_4_m_2_imCl, (**B**) C_2_mimCl, (**C**) C_4_mimCl, (**D**) C_6_mimCl, and, (**E**) from NIST 2684 with C_6_mimCl as the extractant.

**Figure 3 nanomaterials-11-00664-f003:**
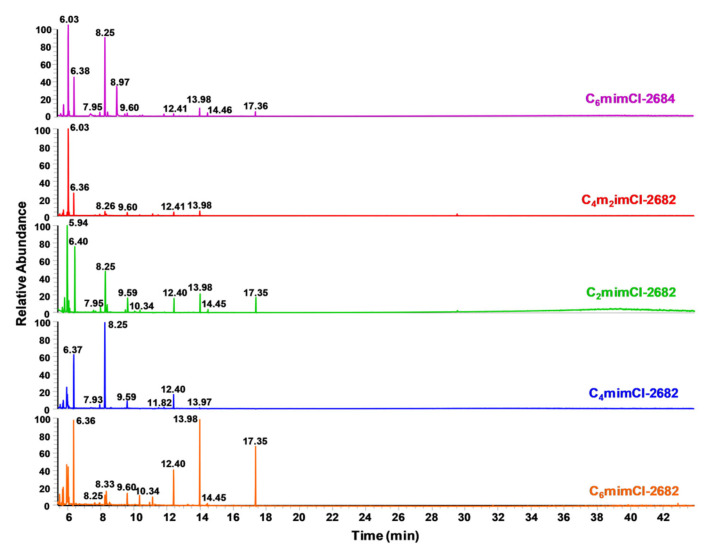
The normalized gas chromatography (GC) retention spectra of ILs-extracted coal precursors.

**Figure 4 nanomaterials-11-00664-f004:**
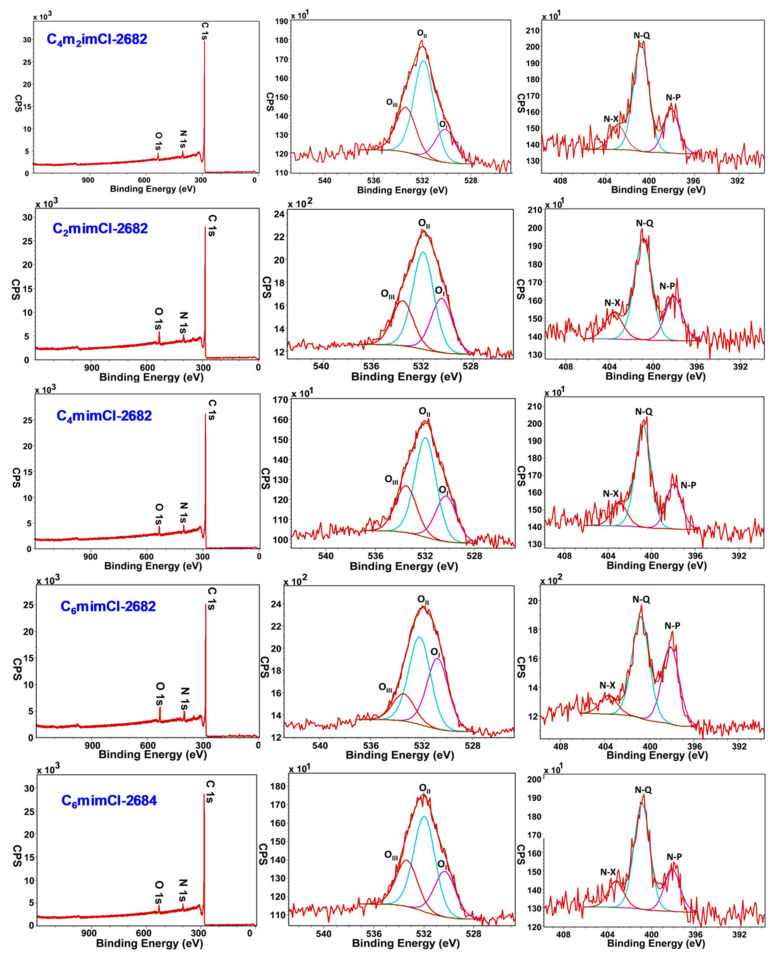
XPS spectra of survey scans (left column), corresponding high resolution scan of O1s (middle column), and N1s (right column) of fabricated CCNFs.

**Figure 5 nanomaterials-11-00664-f005:**
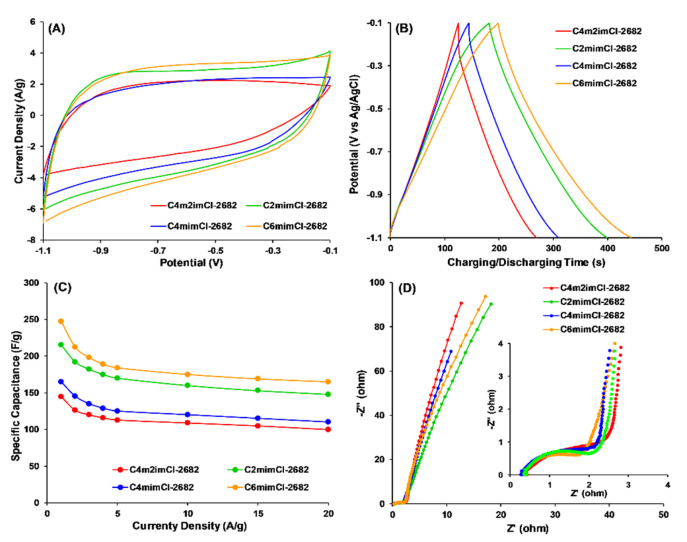
(**A**) Cyclic voltammetry (CV) profiles at 10 mV/s scan rate, (**B**) galvanostatic charge–discharge (GCD) profiles at 1A/g current density; (**C**) rate performances and (**D**) electrochemical impedance spectroscopy profiles of CCNFs fabricated using ILs-extracted 2682 coal precursors.

**Figure 6 nanomaterials-11-00664-f006:**
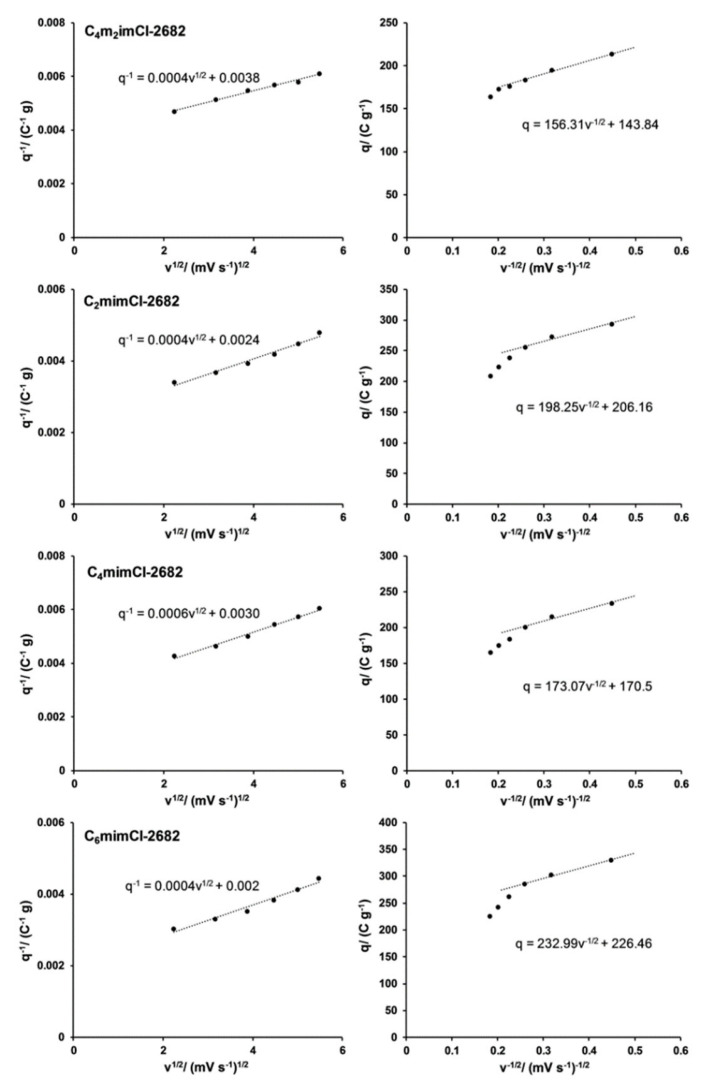
Dependence of 1/q vs. v^1/2^ (left column) and dependence of q vs. v^−1/2^ (right column) of CCNFs fabricated using ILs-extracted 2682 coal precursors.

**Figure 7 nanomaterials-11-00664-f007:**
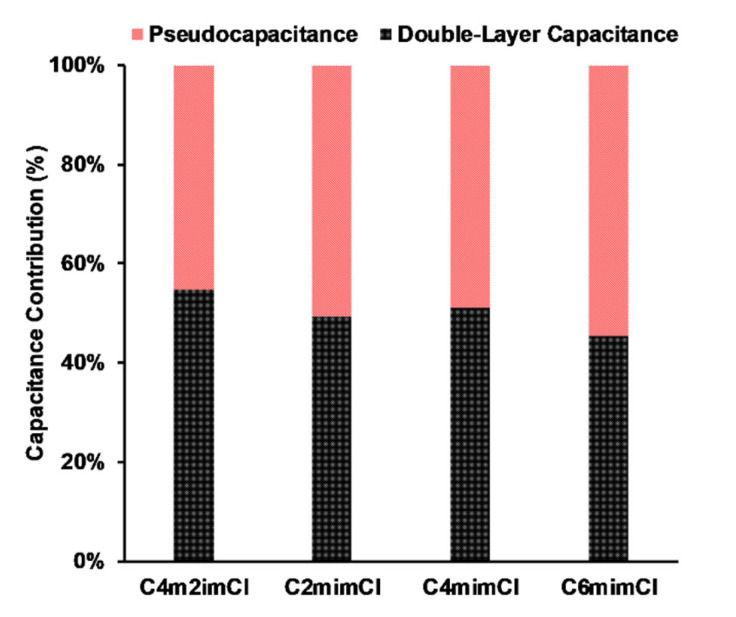
The distribution of C_p_ and C_EDL_ of CCNFs fabricated using ILs-extracted 2682 coal precursors.

**Figure 8 nanomaterials-11-00664-f008:**
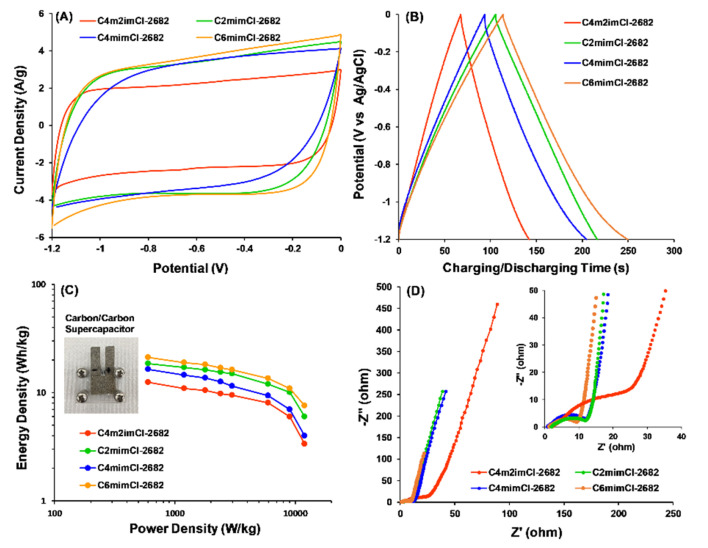
(**A**) Cyclic voltammetry (CV) profiles at 50 mV/s scan rate; (**B**) galvanostatic charge–discharge (GCD) profiles at 1A/g current density; (**C**) relationship between the energy density and power density and (**D**) electrochemical impedance spectroscopy profiles of carbon/carbon symmetry supercapacitor cells using CCNFs as electrodes.

**Figure 9 nanomaterials-11-00664-f009:**
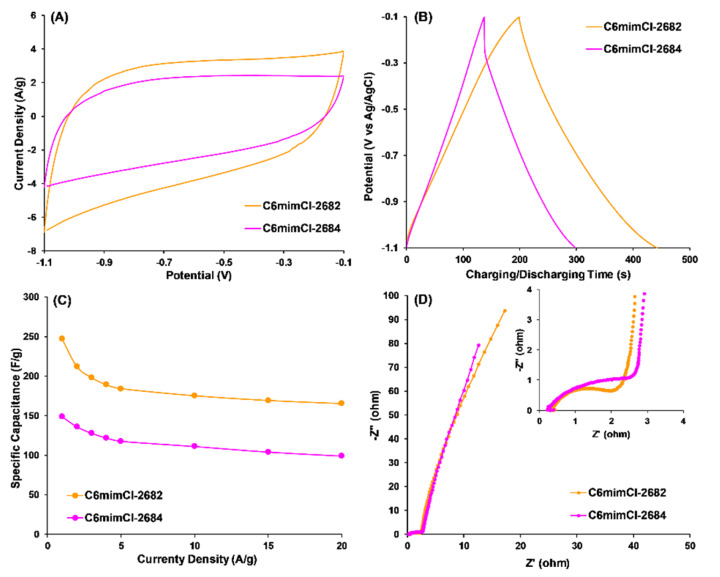
(**A**) Cyclic voltammetry (CV) profiles at 10 mV/s scan rate; (**B**) galvanostatic charge–discharge (GCD) profiles at 1A/g current density; (**C**) rate performances and (**D**) electrochemical impedance spectroscopy profiles of C_6_mimCl-2682 and C_6_mimCl-2684 CCNFs.

**Figure 10 nanomaterials-11-00664-f010:**
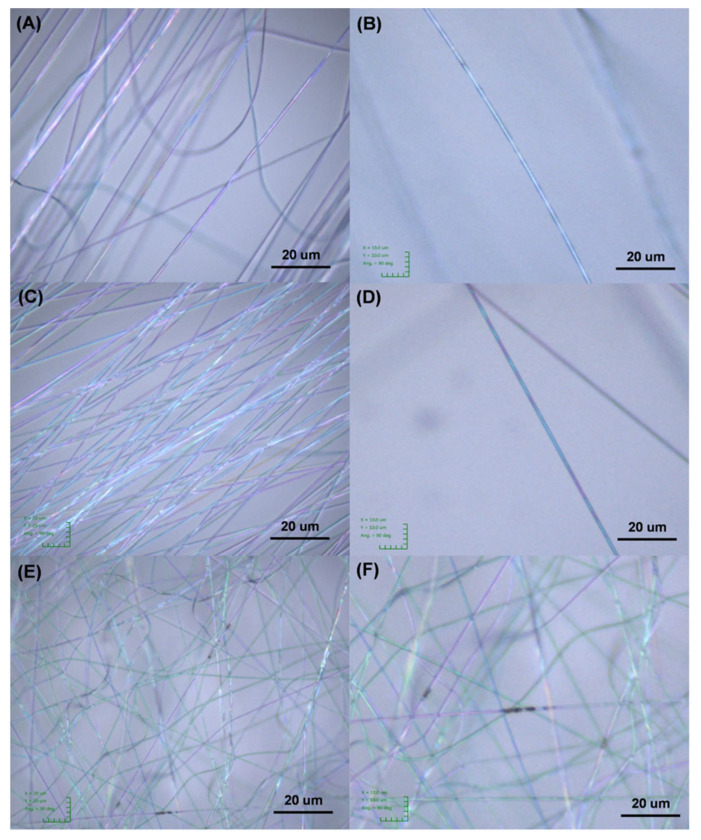
The microscope images showed the homogeneity and heterogeneity of as-spun nanofibers derived from different precursors: (**A**,**B**) poly (acrylonitrile-co-methyl acrylate) (PANMA)/polyvinylpyrrolidone (PVP) precursor; (**C**,**D**) mixture of C_6_mimCl-2682 coal precursor and PANMA/PVP; (**E**,**F**) mixture of C_6_mimCl-2684 coal precursor and PANMA/PVP, indicating the heterogeneity of nanofibers from C_6_mimCl-2684 coal precursor. Note: (**B**,**D**,**F**) are closeups of (**A**,**C**,**E**). The average diameter of all as-spun fibers is 1 µm.

**Table 1 nanomaterials-11-00664-t001:** The distribution of oxygen species obtained by fitting O1s X-ray photoelectron spectroscopy (XPS) spectra, and C, N, O overall concentrations on the surface of fabricated coal-derived carbon nanofibers (CCNFs).

CCNFs	High-Resolution Scan			Survey Scan		
	O 1s%			O%	N%	C%
	O_I_	O_II_	O_III_			
C_4_m_2_imCl-2682	18.9	55.4	25.7	1.9	4.0	94.1
C_2_mimCl-2682	29.5	47.5	23.0	4.1	4.1	91.8
C_4_mimCl-2682	23.8	51.8	24.5	2.6	4.4	93.0
C_6_mimCl-2682	38.1	47.2	14.7	4.2	5.4	90.4
C_6_mimCl-2684	24.3	51.0	24.7	2.6	3.7	93.7

**Table 2 nanomaterials-11-00664-t002:** The summary of total resistance (R_sum_), ion migration resistance (R_s_), internal resistance of electrode (R_ct_) and ion diffusion resistance into pores (R_p_) for fabricated CCNFs.

CCNFs	R_sum_ (Ω)	R_s_ (Ω)	R_ct_ (Ω)	R_p_ (Ω)
C_4_m_2_im-2682	3.985	0.24	3.70	0.045
C_2_mim-2682	2.597	0.26	2.30	0.037
C_4_mim-2682	3.092	0.24	2.80	0.052
C_6_mim-2682	2.275	0.26	1.98	0.035
C_6_mim-2684	3.828	0.24	3.50	0.088

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
