# Peer review of "Investigation on the Mass Distribution and Chemical Compositions of Various Ionic Liquids-Extracted Coal Fragments and Their Effects on the Electrochemical Performance of Coal-Derived Carbon Nanofibers (CCNFs)"

_nanomaterials, 2021, doi:10.3390/nano11030664_

Round 1
Reviewer 1 Report
The publication show interesting results of the materials that can be used for supercapacitors production. The publication may be published after minors corrections.
- The formulas in the text are cut off.
- Authors didn’t explain why C6mimCl-2684 have so low oxygen concentration observed in XPS spectra?
- Authors wrote in Figure 5 caption “impedance spectroscopy profiles of fabricated CCNFs using ILs-extracted 2682 coal precursors” and I think it should be “impedance spectroscopy profiles of CCNFs fabricated using ILs-extracted 2682 coal precursors” (also in other figures).
Author Response
1. The cut-off formulas in Equation 1 to 3 have been fixed.
2. An explanation has been added in lines 289-293 as follows: 'Moreover, due to the low level oxygen contribution from the carbon-rich NIST 2684 bituminous coal and asphaltene feature of the C6mimCl-2684 coal extract (Table.S1), the resulting C6mimCl-2684 CCNFs only contained 2.6% overall oxygen concentration, which was much lower than the 4.2% oxygen content in the C6mimCl-2682 CCNFs.'
3. The captions have been revised in Figure 5 to Figure 7.
Reviewer 2 Report
The only results I do not like are shown in Fig. S1. Such shapes mean that there are a lot of impurities in the samples, moreover in the case of C6mimCl-2682 explosion? is highly endothermic?
Author Response
1. During the oxidation and carbonization of fibers in the TGA analysis, there are lots of steps involved in order to fabricate CCNFs from as-spun fibers, including PVP decomposition, PAN cyclization and carbon ring condensation. Those steps caused the weight loss during the TGA analysis.
We added a couple of sentences in lines 153-161 to address the reviewer's comment as shown below:
'Specifically, in the oxidation step (room temperature to 300oC), nearly 5 wt% water in the as-spun fibers was removed in the early stage (<100oC), following by a 40 wt% mass loss from 200oC to 300oC. Such weight loss was caused by the decomposition of PVP as a sacrificial agent to create the porous structure and by the exothermic process of PAN cyclization. Sequentially, in the carbonization process, carbon ring condensation and breakage of oxygen and nitrogen functional groups occurred, which were usually accompanied with the release of CH4, CO2 and CO. Approximately, 25 wt% weight loss was observed between 400 oC and 1000oC in the carbonization process, which generated CCNFs with oxygen residues in the turbostratic carbon structure.'
Reviewer 3 Report
In this paper, assisted by laser desorption/ionization (LDI) and gas chromatography-mass spectrometry (GC-MS) technologies, a systematic study was performed to characterize mass distribution and chemical composition of coal precursors derived from various ionic liquids (ILs) as extractants. This is an innovative study for the preparation of coal precursors using ionic liquids not organic solvents, and has some valuable contributions in the experimental results, which is worth to appear in the Nanomaterials Journal. The reviewer thinks the manuscript is worth to be published in Nanomaterials Journal, as long as after some minor mandatory amendments or clarifications are done:
- Traditionally, raw coal was depolymerized by using organic solvents, this study used ionic liquids (ILs) as an environmental friendly solvent for extracting coal fragment. This is an interesting and effective study. But please give a comparison between organic solvents and ionic liquids about the quality or yield of prepared coal derivatives.
- The scale bar in Figure 10 is not clear, and the average diameter of the fibers also should be pointed out.
- The supercapacitor fabricated with CCNF as electrode delivered an energy density of 21.1 Wh/kg at the power density of 0.6 kW/kg, please give a table of performance comparison with previous similar studies.
- The conclusion is too lengthy and should be improved.
- The table’s format is disorder in Supplementary Information file.
Author Response
1. Our previous study (Energy Fuels 2020, 34, 4554−4564) performed systematic study on the yield and quality of coal fragments derived from ILs extraction, and compared them with the traditional organic solvents. To address reviewer's comments, we added a couple of sentences to highlight the points in Line 66 to Line 75:
'Most importantly, our following study found that even though the yield of coal fragments derived from ILs extraction (15%-30%) was lower than the one obtained from traditional organic solvents (e.g. 50% using tetralin), the ILs, such as 1-butyl-2,3-dimethylimidazolium chloride (C4m2imCl), can have strong selectivity towards certain groups of fragments in coal extraction, such as a preference to both polycyclic aromatic hydrocarbons (PAHs) and fatty acids (FAs).[50] Therefore, the selectivity in extracting coal compounds is an unique feature of ILs, and it is worth further investigating the role of chemical properties of ILs-derived coal precursors in the electrochemical performances of resulting CCNFs.'
2. A clear scale bar has been added in Figure 10. An average diameter of all as-spun fibers (10 um) was added in the caption of Figure 10. (Line 446)
3. A comparison table (Table.S5) has been added in the supporting information, and briefly discussed in lines 389-390.
4. We deleted the unnecessary paragraph from lines 467-474.
5. The supplementary information was provided in PDF file, which had correct order of the tables.